# Current Understanding of Microbiomes in Cancer Metastasis

**DOI:** 10.3390/cancers15061893

**Published:** 2023-03-22

**Authors:** Jiaqi Liu, Feiyang Luo, Liyan Wen, Zhanyi Zhao, Haitao Sun

**Affiliations:** Clinical Biobank Center, Microbiome Medicine Center, Department of Laboratory Medicine, Zhujiang Hospital, Southern Medical University, Guangzhou 510280, China; 3190017003@i.smu.edu.cn (J.L.); 3210101024@i.smu.edu.cn (F.L.); biobank@smu.edu.cn (L.W.); zhanyi1998@smu.edu.cn (Z.Z.)

**Keywords:** microbiome, cancer metastasis, EMT, immunity, FSS, MMPs

## Abstract

**Simple Summary:**

Microbiomes, inhabiting multiple parts of the human body, are considered to be associated with the progress, treatment, and prognosis of cancer. Metastasis is a main cause for high mortality of cancer, and there is increasing attention to the issue of the relationship between microbiomes and cancer metastasis. This review summarizes the recent advances in this field.

**Abstract:**

Cancer has been the first killer that threatens people’s lives and health. Despite recent improvements in cancer treatment, metastasis continues to be the main reason for death from cancer. The functions of microbiome in cancer metastasis have been studied recently, and it is proved that microbiome can influence tumor metastasis, as well as positive or negative responses to therapy. Here, we summarize the mechanisms of microorganisms affecting cancer metastasis, which include epithelial-mesenchymal transition (EMT), immunity, fluid shear stress (FSS), and matrix metalloproteinases (MMPs). This review will not only give a further understanding of relationship between microbiome and cancer metastasis, but also provide a new perspective for the microbiome’s application in cancer metastasis prevention, early detection, and treatment.

## 1. Introduction

Cancer comprises a bewildering assortment of diseases that kill 7.5 million people each year [1]. Most cancers show improved prognosis recently as a result of advances in cancer therapy. However, there are still some poor prognoses of cancer because most patients are metastasized during diagnosis, and the treatment effect is limited, which is the reason why most cancer-related deaths have occurred [2,3]. Current treatments for cancer include surgical resection, radiation therapy, chemotherapy, and targeted drugs. However, surgical resection and radiation therapy are better for substantial tumors. Although chemotherapy can inhibit tumor cell growth, it cannot inhibit the sowing of tumor cells [4]. Targeted drug treatment can inhibit tumor cell metastasis, but its development is still limited. For example, although immune checkpoint inhibitors have been created to treat metastatic melanoma, 50% of the metastatic melanoma is still invalid for the examination checkpoint inhibitors. Due to the frequent metastasis of the tumor, cancer is still difficult to cure, and the development of effective treatment to suppress its metastasis is imminent.

Cancer metastasis is the dissemination of cancer cells from the primary tumor to other distant organs or tissues [5]. It involves the dissociation of the tumor from the original site, followed by proliferation, entry, and survival in the circulatory or lymphatic system, as well as extravasation and proliferation at the secondary tumor site. These processes allow the cancer cells to evade the host immune system’s monitoring [6]. The primary tumor can communicate with distant organs, converting them into fertile surfaces for circulating tumor cells to implant and multiply, a process known as the pre-metastasis niche (PMN). The PMN [7], which recruits stromal cells to support tumor growth, is just one of the many factors contributing to the complexity of cancer metastasis progression.

About 10^13^~10^14^ microbiota live in the gastrointestinal tract, as well as other tissues that were previously thought to be sterile [8,9,10,11,12,13]. The genetic material carried by these microorganisms is called the microbiome and far exceeds that of the human genome [14]. In the past decade, microbiome communities have been suggested to influence the development, progression, metastasis formation, and treatment response of multiple cancer types [15]. First, several specific microbiota were found to be significantly associated with infection-associated cancers, such as *H. pylori*, *hepatitis B virus*, *hepatitis C virus*, and the human papillomavirus [16,17,18]. Additionally, there is a positive or negative relationship between microbiome and cancer metastasis [19]. Increasing evidence has suggested that gut microbiota can influence cancer spreading by altering the immune system as it accommodates more than half of the body’s lymphocytes [20]. Intestinal dysbiosis-induced sustained liver inflammation can promote cancer development and metastasis [21,22]. The metastasis of colorectal cancer (CRC), oral squamous cell carcinoma, prostate cancer, and pancreatic cancer is also found to be promoted by the microbiome [23,24,25,26]. Interestingly, both microbiome and microbial metabolites can improve cancer treatment by inhibiting cancer metastasis. The possible mechanisms involve the anti-inflammatory potential of the microbiome, which contributes to the inhibition of cancer metastasis [27]. Besides, inhibitors targeting the structure of microbial metabolite can hinder the late and early phases of cancer metastasis [28]. Therefore, as a non-negligible body component, the microbiome plays a crucial mediating role in regulating cancer susceptibility and tumor progression. Additionally, studies in these field are helpful in the development of combination cancer therapies. In this review, we will summarize the potential roles and mechanisms of microbes affecting cancer metastasis.

## 2. The Roles of Microbiome in Cancer Metastasis

Tumor cells with high malignancy can initiate metastatic foci and probably result in fatal effects on patients. The survival of these pernicious cancer cells can be improved by promoting cell viability through epigenetic states (e.g., EMT), microenvironmental factors (e.g., cancer-activated immune cells), physical factors (e.g., FSS), or genetic alterations (e.g., DDR). Microbiomes, such as *Fusobacterium nucleatum (Fn)*, *Porphyromonas gingivalis*, and *Akkermansia*, as well as microbial metabolites, such as indolepropionic acid (IPA), cadaveric amines, and sodium butyrate (NaB), play critical roles in the metastasis of breast cancer, melanoma, prostate cancer (PC), and CRC via multiple mechanisms [29,30,31,32,33]. (1) The microbiome can induce EMT and promote cancer metastasis through decreasing β-linked proteins and activating the Wnt signaling pathway [34,35,36]. Additionally, microbiota upregulate E-cadherin expression by multiple mechanisms, inhibiting the EMT process [32]. (2) The correlation between microbial abundance and immune cell populations was analyzed, and a significant relationship between microbial abundance and immunity was found. Immune cells, such as macrophages with M2 polarization, and the T helper 17 (Th17) cells, are implicated in cancer metastasis [37,38,39]. (3) The microbiome can help tumor cells adapt to the biochemical and biophysical factors in the tumor microenvironment during the metastatic process. For example, shear stress induced by fluid flow as a classical physical factor contributes to the metastasis of tumor cells [40]. Additionally, the microbiome has been shown to alter the actin cytoskeleton in a way that increases the resistance of tumor cells to FSS [4]. (4) Microbiome selectively increased MMP-1/9/10 and MMP-2F/9F, while the former acted to promote cancer metastasis [30,41], and the latter represses it [42]. (5) The DNA damage repair (DDR) system is essential to maintain genome integrity after excessive damage types and for cancer cell survival [43]. Additionally, it has been proven that the microbiome can affect DDR [44]. Admittedly, there is a lack of specific experiments to prove that microbiota affects cancer metastasis by influencing DDR, but its potential connection is worth further investigation. These results are summarized in Table 1 and Table 2.

## 3. Microbiome Influences Cancer Metastasis through Epithelial-Mesenchymal Transition (EMT)

Epithelial-mesenchymal transition (EMT) refers to a process of cellular reprogrammed ability wherein epithelial cells lose their adhesion to neighboring cells and the extracellular matrix (ECM) [59], and then they develop mesenchymal characteristics required for invasion and migration at the same time [60]. It occurs when cancer cells become metastatic cells and is characterized by a crucial molecular and cytomorphological transition. EMT links to tumor invasion, metastasis, and drug resistance [59]. To develop an invasive phenotype for metastatic progression in cancer, carcinoma cells use EMT to speed up their dissociation from primary tumors and their dissemination into blood circulation [61].

Recently, it has been found that microbiota can influence cancer metastasis by inhibiting or facilitating EMT. For example, Cheng Kong et al. [45] identified that *Fn* promoted EMT and metastasis in CRC by activating a TLR4/Keap1/NRF2 axis to increase CYP2J2 and its metabolite 12,13-EpOME. In individuals with CRC, metagenomic sequencing revealed a significant link between elevated fecal *Fn* and blood 12,13-EpOME levels. Elevated levels of CYP2J2 in tumor tissues were similarly associated with high *Fn* levels and poor overall survival in patients with stage III/IV CRC. This study highlights the key role of *Fn* in altering tumor cell metabolism in the field of CRC, while metabolomics is at the forefront of histological approaches that allow qualitative and quantitative monitoring of metabolomic changes and provide new directions for cancer metastasis surveillance. However, *Fn* can not only promote EMT by altering tumor cell metabolism leading to metastasis, but also increases miR-155-5p and miR-205-5p expression through activation of innate immune signaling to suppress alcohol dehydrogenase 1B (ADH1B) and transforming growth factor β receptor 2 (TGFBR2) expression. Upregulation of both microRNAs leads to reprogramming of ethanol metabolism, allowing *Fn* accumulation and activation of the PI3K/AKT148 signaling pathway to promote EMT. Additionally, EMT is a key step in the invasion and metastasis of laryngeal squamous cell cancer (LSCC) and a major cause of poor prognosis. That is the positive feed-forward loop between *Fn* and ethanol metabolism reprogramming that promotes LSCC metastasis [46]. Moreover, a typical change in EMT is the loss of E-calmodulin accompanied with dysregulation of the Wnt signaling pathway [34]. E-calmodulin is a tumor suppressor that acts through β-linked protein [35,36]. Mara et al. found that *Fn* can bind and induce phosphorylation/internalization of E-calmodulin and decrease β-linked proteins. Additionally, it can also further activate the Wnt signaling pathway (phosphorylation/degradation of GSK3β and breakdown of the APC/Axin/GSK3β complex) by its FadA adhesin binding, leading to enhanced EMT and invasion of CRC cells [29]. In addition to this, *P. burgdorferi* has been reported to induce EMT through the downregulation of nucleoplasmic accumulation of E-calmodulin and β-linked protein, thereby inducing aggressive and/or metastatic potential of oral squamous cell carcinoma (OSCC) [47].

Not only can the microbiota affect EMT, but studies have shown that some microbial metabolites can also affect EMT and thus alter the metastatic nature or number of cancer stem cells, thereby affecting cancer metastasis. EMT gives tumor cells more stemness and improved resistance to immune eradication and different therapeutic assaults [62]. For example, the supplementation of IPA [32], a tryptophan metabolite produced only by intestinal flora [63], reduced the proportion of cancer stem cells and the proliferation, motility, and metastasis formation of cancer cells. These were achieved by inducing the expression of mesenchymal markers, vimentin (Vim), fibroblast growth factor binding protein 1 (FgfBp1), snail family transcriptional blocker-1 (Snail), and β-linked protein; this was also accomplished by upregulating the expression of E-calmodulin, i.e., inhibiting EMT to suppress breast cancer metastasis. In addition, cadaveric amines produced by the intestinal microbiota reduce the motility and metastatic nature of cancer stem cells by suppressing EMT, thus exerting a tumor-suppressive effect in breast cancer [54]. There are a number of microbiota whose presence is associated with the expression of EMT-related genes, but their specific pathways remain to be explored. Jin-Yong Jeong et al. [48] found that the presence of *T epidimonas fonticaldi (TF)* supernatant induces cancer cell proliferation and migration, EMT-related mRNA, and TCA cycle-related metabolites, which are significantly increased. Therefore, it is reasonable to speculate that it promotes lymph node metastasis in pancreatic ductal adenocarcinoma (PDAC) by promoting EMT. Furthermore, by correlating the microbial abundance of tissue specimens from patients with muscle-invasive bladder cancer (MIBC) with the expression of EMT-related genes, Li et al. [55] found that the presence of *Oscillatoria* was strongly negatively correlated with EMT-promoting genes, presumably inhibiting metastasis in MIBC.

These studies have found that EMT is an important medium between the tumor microbiome and cancer metastasis, and the relevant mechanism is explained. The signal pathway involved in this process may be an effective treatment target for cancer metastasis, which indicates that the microbiome has great potential in cancer treatment, summarized in Figure 1.

## 4. Microbiome Influences Cancer Metastasis by Modulating Immunity

The innate and acquired immune systems have a significant impact on cancer occurrence, and they can encourage or inhibit tumor metastasis. Changes in the development or composition of the microbiome (ecological dysbiosis) may interfere with the partnership between the microbiome and the human immune system, ultimately leading to altered immune responses that may underlie various human inflammatory diseases [51]. Inflammation is universally regarded as a characteristic of innate immunity. Activation of innate immunity by foreign microbial and viral structures results in the upregulation of MHC class I and II and co-stimulatory molecules, as well as numerous inflammatory chemokines and cytokines. Additionally, they draw and cause NK-cell and T-cell activation through various antigen receptors [37]. However, tumor cells have developed a range of immune escape strategies that have reduced immune cell effector functions.

A large number of studies have shown that the microbiota affect cancer metastasis by modulating immunity, thereby assisting or inhibiting immune escape from tumour cells. For example, Parhi et al. [10] found that *Fn* colonizes breast tumors and causes a decrease in CD4+ and CD8+ T cells through the abundant Gal-GalNAc (also known as Thomsen-Friedenreich antigen) on tumor cells, so it is reasonable to speculate that *Fn* not only inhibits NK cells and tumor-infiltrating T cells killing of cancer cells [64], but also inhibits the accumulation of tumor-infiltrating T cells and promotes tumor growth and metastatic progression. In addition to this, the presence of *Fn* was associated with a lower density of CD8+ T cells and a higher density of bone marrow-derived suppressor cell (MDSC) markers in colorectal cancer liver metastasis (CRCLM) tissues [49]. This suggests that the microbiota can assist tumor cells in downregulating MHC class I on the cell surface, thereby reducing antigen-dependent activation of T cells and elimination of cytotoxic CD8+ T lymphocytes (CTL). Not only can *Fn* suppress immunity to promote cancer metastasis, but *Staphylococcus aureus* has also been reported to be highly associated with the expression of regulatory T cells in PC, which inhibit the activation and proliferation of effector T cells and weaken the immune system. It is suggested that they drive PC metastasis by inducing inflammation, promoting immune suppression, and regulating prostate cancer stem cells (PCSCs) expression [25]. The stimulation of stem cell development and survival pathways, as well as immunological defenses such as cytotoxic T cells and natural killer T (NKT) cells, boost the metastatic proliferation of tumor cells [56], whereas the microbiota compromise the immune system, encouraging cancer metastasis. This has also been demonstrated in gut microbes. Subhashis Pal et al. [50] demonstrated that gut microbes can inhibit the growth of bone marrow NK and Th1 cells by blocking the S1P-S1PR1/5 axis and CXCR3-CXCL9 chemokine gradient. They also discovered that gut microbes can promote the expression of inflammatory cytokines in NK cells and upregulate the expression of chemokine receptors, thus facilitating their migration to distant organs [65] and accelerating the bone metastasis of melanoma.

Notably, recent studies have found that metabolites of the microbiome inhibit cancer metastasis by increasing the number of T helper 17 (Th17) cells. For example, sodium butyrate (NaB) [56], a fermentation product of intestinal microorganisms, effectively regulated the intestinal flora of CRLM mice, increasing the proportion of *Bacteroidetes* and decreasing the proportion of *Firmicutes*, decreasing the number of liver T regulatory cells in CLM mice and increasing the number of NKT cells and Th17 cells, thereby decreasing IL-10 levels and increasing IL-17 secretion. The Th17 /T regulatory cell (Tregs) ratio and cytokine imbalance are important for CRC development [39]. Therefore, it is reasonable to speculate that NaB suppresses liver metastasis in CRC by improving immunity. Moreover, Chen et al. [57] found that propionic acid and butyric acid produced by intestinal flora after probiotic supplementation promoted the expression of chemokine (C-C motif) ligand 20 (CCL20) in lung endothelial cells. Additionally, they also attenuated pulmonary metastasis of mouse melanoma cells by facilitating the recruitment of Th17 cells to the lung via the CCL20/chemokine receptor 6 axis. Interleukin-17 (IL-17)-producing Th (Th17) cells are categorized as an inflammatory Th subpopulation within the Th subpopulation, which results in chronic tissue inflammation and ultimately organ failure [66]. Numerous human malignancies have been shown to contain Th17 cells. These cells exhibit both tumor-promoting and tumor-suppressing activity, and their role in malignancies is very context-dependent [67]. Because of its complexity, cancer therapies targeting Th17 are mostly in the experimental stage, and the above microbial metabolites and their signaling pathways may provide new ideas.

In addition, it has been noted that tumor-associated macrophages (TAMs) play a supporting role in the onset and spread of cancer by promoting cell survival, proliferation, angiogenesis, invasive and motile behavior, and inhibiting CTL response at primary and metastatic sites [38]. Macrophages can polarize macrophages into M1 or M2 phenotypes [68]. M2 phenotype macrophages in the tumour microenvironment (TME) tend to exert an immunosuppressive phenotype, and microbial promotion of M2 polarization of macrophages plays a crucial role in promoting tumor progression. Li et al. [23] found that the *E. coli* group, which also had higher levels of LPS release and the metastasis-associated secretory protein cathepsin K(CTSK) overexpression, had more liver metastases and larger primary tumors than the controls. Additionally, CTSK could stimulate the M2 polarization of TAMs through a mTOR-dependent pathway, a key intracellular signaling system that drives cell growth and survival. The M2 polarization of TAMs induces the secretion of cytokines, including IL10 and IL17, which in turn promoted the invasive metastasis of CRC cells through the NF-κβ pathway. Interestingly, Xu et al. [68] revealed that *Fn* decreased the expression of miR-1322 in CRC cells by triggering the NF-κβ signaling pathway. The chemokine CCL20, the sole known ligand for the receptor CCR6, can be directly bound by miR-1322, a putative regulatory micro RNA. This interaction not only influences cancer growth, but it also promotes CRC spread by affecting immune cells participating in TME reprogramming [69]. Activated tumor-derived CCL20 promotes macrophage infiltration while inducing M2 macrophage polarization and enhancing CRC metastasis. According to reports [52], *Trichomonas vaginalis* infection promotes inflammation by secreting the pro-inflammatory cytokine IL-6, which drives macrophages to become M2 polarized. Additionally, the studies of Ik-Hwan Han et al. show more details. They demonstrate that a conditioned medium containing IL-6-containing prostate epithelial cells co-cultured with *Trichomonas vaginalis* promoted M2 polarization of macrophages by upregulating M2 markers, such as IL-10, TGF-β, CD36, CD206, and arginase-1 and that M2 polarization in THP-1-derived macrophages is regulated by the IL-6R/JAK signaling pathway. This encouraged the migration and proliferation of PC cells. The aforementioned microbiota can promote M2 polarization of macrophages through various signaling pathways, and M2-like macrophages assist in metastasis, angiogenesis, and proliferation of cancer cells through various anti-inflammatory mechanisms.

Overall, there is accumulating evidence that the microbiome affects systemic inflammation and immunity. Additionally, there are multiple possible mechanisms linking microbiome to carcinogenesis, tumor outgrowth and metastases, altered metabolism, pro-inflammatory, and impaired immune-response. Through the study of microbes and their pathways, immunotherapy associated with them can be an effective approach to inhibit cancer metastasis, summarized in Figure 2.

## 5. Microbiome Affects Cancer Metastasis by Influencing Fluid Shear Stress (FSS)

The application of microfluidics and mechanical measurement in tumor research is becoming more and more extensive, which has accelerated the development of tumor hydrodynamics. There is mounting evidence that FSS plays a significant role in metastasis and that it has a great impact on hydrodynamics [40].

Metastatic cancer cells undergo FSS, which frequently results in apoptosis, especially following venous entrance into the circulatory system [70]. Under physiological circumstances, intracellular bacteria can circulate in the bloodstream with cancer cells and contribute significantly to metastatic colonization. According to research by Fu et al. [4], *S. xylosus*, L. *animalis*, and *S. cuniculi* alter the actin cytoskeleton in a way that increases tumor cells’ resistance to FSS. This increases host cell survival and thus breast cancer of lung metastasis. In particular, interference with intracellular bacteria reduces metastasis, but not primary tumor growth.

## 6. Microbiome Influences Cancer Metastasis by Regulating Matrix Metalloproteinases (MMPs)

Matrix metalloproteinases (MMPs) are zinc (Zn2+)-dependent endopeptidases that are present intracellularly and membrane-bound. The majority of the 26 MMPs that have been discovered are found in the human proteome [71]. Members of the MMPs family can be classified into several classes, such as gelatinases, collagenases, stromelysins, stromelysins, and membrane-type matrix metalloproteinases [72]. Uncontrolled tumor growth, local invasion, and metastatic cancer progression are largely dependent on the proteolytic activity of many MMPs. They affect tissue integrity, immune cell recruitment, and tissue renewal by degrading ECM components and releasing matrix factors, cell surface-bound cytokines, growth factors, or their receptors [73]. Thus, MMPs mainly affect the steps in cancer metastasis that separate from the primary tumor site and enter and survive in the circulatory or lymphatic system.

With the increase in related research reports, studies on microbial influence on cancer metastasis through regulation of MMPs have become a research hotspot, in which MMP-1, MMP10, MMP-9, and MMP2 have all been shown to act as important regulators between the microbiome and cancer metastasis, summarized in Figure 3. MMP-1 and MMP-10 primarily affect tissue integrity by degrading ECM components, thereby promoting cancer metastasis. Ha et al. [41] found that *Porphyromonas gingivalis* can stimulate the secretion of MMP-1 and MMP-10 through the release of IL-8 and gingival protease, thus increasing the invasiveness of cancer cells. MMP-2 and MMP-9 are also gelatinases capable of degrading and modifying the ECM [74]. Yue et al. [58] found that metabolite secretion of *Lactobacillus plantarum YYC-3* may inhibit colon cancer cell metastasis by suppressing the VEGF-MMP2/9 signaling pathway, and in Caco-2 cells, MMP-2, MMP-9, and VEGFA gene expression was significantly reduced in the YYC-3-treated group. In the VEGF-MMPs signaling pathway, VEGF binds its receptor VEGFR, which is associated with the secretion of downstream target MMPs, thereby inhibiting cancer metastasis. In addition, serum carcinoembryonic antigen (CEA), CEAM6, MMP-2F, and MMP-9F gene expression were reported to be significantly downregulated after lactacystin treatment. Additionally, MMP-2F expression was associated with metastatic ability and poor prognosis, thereby inhibiting lymph node metastasis in colon cancer [42]. In particular, MMP-9 not only affects the step of separation from the primary tumor site in cancer metastasis, but since its activity is also associated with cancer pathology, including invasion, angiogenesis, and metastasis [10], it can establish a metastatic ecological niche after tumor cell extravasation, allowing tumor cells to grow in an unfavorable environment [75]. Adi Binder Gallimidi et al. [30] proved that oral infections cause TLR signaling, which in turn causes IL-6 production and STAT3 activation. In both clinical and experimental settings, STAT3 has a significant role as a modulator of OSCC carcinogenesis [76]. These essential effectors include cyclinD1, MMP-9, and heparinase, which promote the growth and invasion of OSCC. Furthermore, it has been reported that *EBV* promotes cancer cell invasion and tumorigenesis by upregulating the expression of cancer stem cell markers through the mechanism of CTAR family proteins upregulating programmed cell death protein 1 ligand (PD-L1); this results in decreasing the stability of p53 and increasing the secretion of MMPs [53].

MMPs play an important role in many biological processes involving matrix remodeling, and a large body of experimental and clinical evidence also suggests that MMPs are associated with tumor invasion, neovascularization, and metastasis, which may be the ideal pharmacological target for tumor treatment. By exploiting the microbiome or its pathways, it may be possible to contain MMPs at the source and thus aid in the treatment of cancer metastasis.

## 7. Discussion

To offer relevant researchers a comprehensive understanding of the field, this review summarizes the impact of microorganisms on cancer metastasis, focusing on their specific mechanisms, including EMT, immunity, FFS, and MMPs. Additionally, the importance of DNA damage repair (DDR) in preserving genomic integrity is highlighted, as accumulation of DNA damage or flaws in the DDR system may encourage tumor metastasis and offer potential targets for cancer therapy [77]. Recent research has demonstrated that microbiome can affect DDR. For instance, staphylococcal DNA repair is crucial for the survival of infections in host tissues and encourages the creation of variants that are resistant to both host defense mechanisms and medications [44]. Additionally, many bacterias’ adaptive responses to alkylation damage trigger alkylation damage repair (the Ada response). For instance, in *E. coli*, this is mediated by the N-terminal structural domain of the Ada protein (N-Ada), which is activated by DNA methylphosphate damage to induce transcription of Ada-responsive genes. The same as *E. coli*, *Bacillus subtilis* responds to Ada in a similar manner [78]. Therefore, although there is currently no clear evidence for this, it is reasonable to speculate that the microbiome may be able to disrupt DDR and thus influence cancer metastasis. The potential contribution of microbial studies related to DDR in the treatment of cancer metastasis will offer fresh perspectives on targeted tumor therapy and facilitate the development of new research directions and new strategies [79].

Cancer is a complicated illness that encompasses more than 100 different diseases with numerous subtypes. Despite significant advancements in the fight against cancer, many of the medicines used today are extremely toxic and cause negative side effects for patients [42]. Moreover, some cancers, such as CRC, can remain asymptomatic even if metastasis has occurred, leading to delayed treatment. Therefore, exploring novel ways to predict and treat cancer metastasis is warranted. The aforementioned findings offer a theoretical framework for further investigation into the utilization of the microbiome to create novel approaches for the detection and treatment of patients with metastatic cancer.

However, the limitations of current studies cannot be ignored. To start, the available data support that the gut microbiome is influenced by both extrinsic (e.g., diet, drugs) and intrinsic (e.g., somatic and epigenetic genetic variation, fecal water content, immune response, co-morbidities) host factors, which are routinely obtained or analyzed in many microbiome studies [80]. Furthermore, the annotation of macrogenomic data from microbial taxa is limited to the genus or species level and lacks detail. Additionally, databases dedicated to the annotation of microbial metabolites are scarce. Thus, more extensive and multi-omics investigations are required to fully comprehend the function of microbiota in cancer onset and progression.

This discipline, though, is quickly developing and holds promise for clinical translation. Current research focuses on developing targeted therapies that regulate microbiota mechanisms involved in cancer metastasis, such as fecal microbiota transplantation (FMT) [81], prebiotics [82], dietary changes [83], and antibiotics [84]. Recent studies have shown that caloric restriction (CR) can inhibit tumor growth in mice through a mechanism dependent on intestinal flora, but this inhibition is diminished in the presence of antibiotic depletion of intestinal flora, while *bifidobacteria* supplementation can restore the antitumor effects of CR, significantly inhibiting primary tumor growth and reducing metastatic burden [85]. These findings suggest that the microbiome may serve as a biomarker for cancer metastasis and offer potential therapeutic approaches for clinical application.

## 8. Conclusions

In conclusion, this review covers the elements of microorganisms affecting cancer metastasis and their unique mechanisms, including EMT, immunology, MMPs, the GVB, etc., to give pertinent researchers a comprehensive understanding of the area. It is anticipated that more widespread standardized specimen collection and preservation, more uniform and rational clinical design, and more robust database sharing and sequencing analysis technologies would enable the clinical application of microbiome as cancer metastasis biomarkers and even therapeutic approaches.

## Figures and Tables

**Figure 1 cancers-15-01893-f001:**
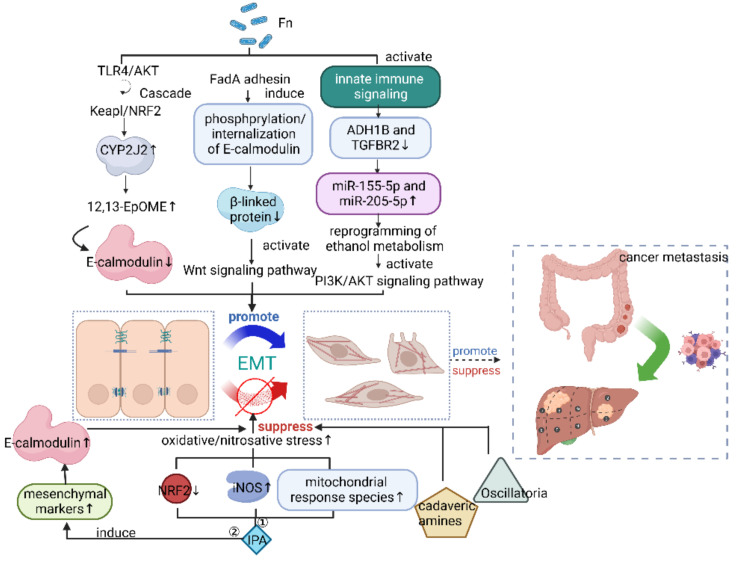
Microbiome affects cancer metastasis through epithelial–mesenchymal transition (EMT). *Fn*, *Fusobacterium nucleatum*; ADH1B, alcohol dehydrogenase 1B; TGFBR2, transforming growth factor β receptor 2; IPA, indolepropionic acid. Figure created with BioRender.com on 14 January 2023.

**Figure 2 cancers-15-01893-f002:**
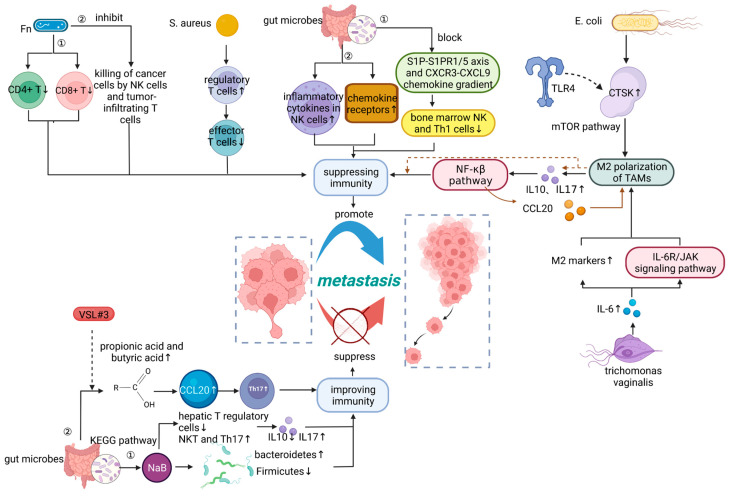
Microbiome influences cancer metastasis by modulating immunity. NK cells, natural killer cells; TLR4, Toll-like receptor 4; CTSK, cathepsin K; Th17 cells, T helper 17 cells; CCL20, chemokine (C-C motif) ligand 20. Figure created with BioRender.com on 14 January 2023.

**Figure 3 cancers-15-01893-f003:**
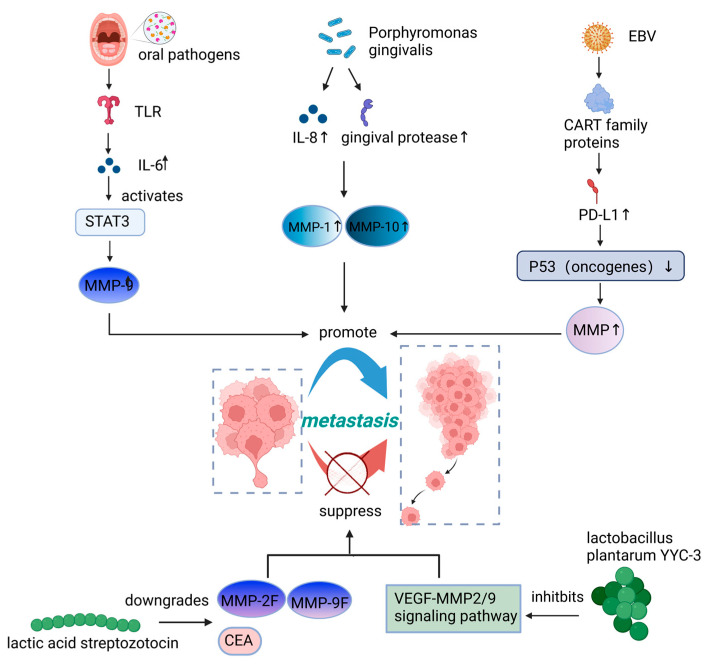
Microbiome affects cancer metastasis by regulating Matrix metalloproteinases (MMPs). PD-L1, programmed cell death protein 1 ligand; CEA, carcinoembryonic antigen. Figure created with BioRender.com on 14 January 2023.

**Table 1 cancers-15-01893-t001:** Summary of pro-metastatic microbiome findings.

Microbiome or Microbial Metabolites	Cancer Type	Impact on Metastasis *	Mechanism	In Vivo/In Vitro	Refs.
*Fusobacterium nucleatum*	CRC	↑	Increases CYP2J2 and 12,13-EpOME (oncogenic metabolites) by activating the TLR4/Keap1/NRF2 axis, thereby promoting EMT	Both	[45]
*Fusobacterium nucleatum*	CRC	↑	Modulates E-Cadherin/β-Catenin signaling via its FadA adhesin and further activates the Wnt signaling pathway, leading to enhanced EMT	Both	[29]
*Fusobacterium nucleatum*	laryngeal squamous cell cancer (LSCC)	↑	Increases miR-155-5p and miR-205-5p expression to suppress ADH1B and TGFBR2 expression, leading to reprogramming of ethanol metabolism to allow Fn accumulation and PI3K/AKT signaling pathway activation to promote EMT	Both	[46]
*Porphyromonas gingivalis*	oral squamous cell carcinoma (OSCC)	↑	Downregulates nucleoplasmic accumulation of E-calmodulin and β-linked protein to promote EMT	In vitro	[47]
*T epidimonas fonticaldi*	pancreatic ductal adenocarcinoma (PDAC)	↑	EMT-related mRNA and TCA cycle-related metabolites are significantly increased	In vitro	[48]
*Fusobacterium nucleatum*	Breast cancer	↑	Inhibits the killing of cancer cells by NK cells and tumor-infiltrating T cells and the accumulation of tumor-infiltrating T cells	Both	[10]
*Fusobacterium nucleatum*	CRC	↑	Lowers the density of CD8+ T cells and increases the density of MDSCs	In vitro	[49]
*Staphylococcus aureus*	PC	↑	Activates regulatory T cells, which suppress the activation and proliferation of effector T cells and impair the immune system	In vitro	[25]
gut microbes	melanoma	↑	Inhibits the growth of bone marrow NK and Th1 cells by blocking the S1P-S1PR1/5 axis and CXCR3-CXCL9 chemokine gradient	In vivo	[50]
*Escherichia coli*	CRC	↑	Causes CTSK overexpression, TLR4, to stimulate M2 polarization of TAMs and secretion of cytokines, including IL10 and IL17 through the motor-dependent pathway, which, in turn, promotes invasive metastasis of CRC cells through the NF-κβ pathway	Both	[25]
*Fusobacterium nucleatum*	CRC	↑	Promotes CRC metastasis through miR-1322/CCL20 axisand M2 polarization	Both	[51]
*Trichomonas vaginalis*	PC	↑	Secretes the pro-inflammatory cytokine IL-6, which drives M2 polarization	In vitro	[52]
*Staphylococcus xylosus*, *Lactobacillus animalis*, and *Streptococcus cuniculi*	breast cancer	↑	Enhances resistance to FSS by reorganizing the actin cytoskeleton	Both	[4]
oral pathogens	OSCC	↑	Causes TLR signaling, which in turn causes IL-6 production and STAT3 activation. Then, it activates essential effectors, including cyclinD1, MMP-9, and heparinase	Both	[30]
*Porphyromonas gingivalis*	OSCC	↑	Stimulates MMP-1 and MMP-10 through the release of IL-8 and gingival protease	In vitro	[41]
*Epstein-Barr virus*	Head and neck squamous cell carcinomas	↑	Decreases the stability of p53 and increases the secretion of MMPs	In vitro	[53]

***** ↑ microbes promote cancer metastasis.

**Table 2 cancers-15-01893-t002:** Summary of anti-metastatic microbiome findings.

Microbiome or Microbial Metabolites	Cancer Type	Impact on Metastasis *	Mechanism	In Vivo/In Vitro	Refs.
Indolepropionic acid (a tryptophan metabolite produced only by intestinal flora)	breast cancer	↓	Induces the expression of Vim, FgfBp1, Snail, and β-catenin; and it upregulates the expression of E-cadherin to suppress EMT	Both	[32]
Cadaverine (produced by the intestinal microbiome)	breast cancer	↓	Reduces the motility and metastatic nature of cancer stem cells by restoring EMT	Both	[54]
*Oscillatoria*	muscle-invasive bladder cancer (MIBC)	↓	Is strongly negatively correlated with EMT-promoting genes	In vitro	[55]
Sodium butyrate (a fermentation product of intestinal microorganisms)	CRC	↓	Decreases the number of hepatic T regulatory cells and increased the number of NKT cells and Th17 cells	In vivo	[56]
Propionic acid and butyric acid (produced by intestinal flora)	melanoma	↓	Facilitates the recruitment of Th17 cells to the lung via the CCL20/chemokine receptor 6 axis	In vivo	[57]
*Lactic acid streptozotocin*	Colon cancer	↓	Downregulates the serum carcinoembryonic antigen (CEA), CEAM6, MMP-2F, and MMP-9F gene expression	In vitro	[42]
*Lactobacillus plantarum YYC-3*	Colon cancer	↓	Suppresses the VEGF-MMP2/9 signaling pathway	In vitro	[58]

* ↓ microbes inhibit cancer metastasis.

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
