# Peer review of "Current Understanding of Microbiomes in Cancer Metastasis"

_cancers, 2023, doi:10.3390/cancers15061893_

Round 1

Reviewer 1 Report

The manuscript entitled "Current understanding of microbiome in cancer metastasis" by Liu and colleagues addresses a very important topic around the key role of microbiome in cancer pathology and response to therapy. This role was highlighted by rcongising the microbiome as a prospective hallmark of cancer.

The manuscript addresses the role of microbiome in cancer metastasis by affecting EMT, immunity, MMPs, FSS and DDR. Despite the interesting topic addressed, there is one major issue: discussing DDR, the authors failed to provide any clear link between the microbiome and DNA damage repair in host cells or tumor cells. They are merely speculating about this potential link. Since there is no clear evidence yet, I would suggest to highlight this speculation in the discussion section, rather than having it as a main pragraph in the paper.
Other minor comments:

1- In the table, please state the full name of the bacterial species and metabolites rather than using acronyms.

2- The term "microbiome" should be used appropriately and not just to refer to single bacterial species.

3- In line 44/45, "targeted drugs" are mentioned. Could you please clarify this?

4- For the sentence in lines 77-79, pleae provide a reference.

I highly recommend going through the manuscript and rephrase/rewrite some long or ambiguous sentences.

Reviewer 2 Report

The article presented by Jiaqi Liu and collaborates, entitled “Current understanding of microbiome in cancer metastasis”, is a review that aimed to summarise the factors of microorganisms affecting cancer metastasis (epithelial-mesenchymal transition (EMT), immunity, fluid shear stress (FSS), DNA damage repair (DDR) and matrix metalloproteinases (MMPs)). The microbiome and its relationship to cancer is a hot topic that a Pubmed search yields 4775 reviews from 2013 to the present. Highlighting the interest in the scientific community. If we use the keywords cancer metastasis and microbiota, 360 results appear, of which 162 are reviews, highlighting the high number of reviews with respect to the original works. The review carried out by Jiaqui Liu is a global review dealing mainly with colorectal cancer, LSCC and OSCC.

Major revision:

1.      The simple summary contains errors that limit the comprehensibility of the text and should be corrected as follows: “The presence and dysbiosis…” “relevant researchers”

2.      The text referred to in reference 13 should be expanded and explained because it is not clear from the text what the objective is. “Between 10 and 100 trillion microbiota live inside the gastrointestinal tract, organs, and tissues that were previously thought to be sterile[8-12], according to estimates[13].”

3.      Meaning NaB in the line “Microbiome such as Fusobacterium Nucleatum (Fn), Porphyromonas gingivalis, Akker mansia, and gut microbes or microbial metabolites comprising indolepropionic acid (IPA), cadaveric amines, and NaB play an important role in cancer metastasis of breast càncer”

4.      In paragraph 2 " The roles of microbiome in cancer metastasis", the relationship between the microbiome and metastatic cancer is not well explained. Some points are related to the microbiome but others are not, for example point 4.

5.      Table one should be divided in two: pro-metastatic microbiome and another table with the anti-metastatic microbiome findings.

6.      NK line 201

7.      Authors should show what their databases have been, as well as how they have searched for information (keywords, inclusion criteria, etc.).

Round 2

Reviewer 2 Report

After reading the paper and responses provided by the authors of the review titled "Current understanding of microbiome in cancer metastasis". I have verified that the authors have introduced the suggested changes in the work and have answered each of my questions justifying the answers in a coherent way. For all these reasons, I consider that the work meets the requirements to be published.